# Cost efficiency of commercial banks in Ethiopia: Does financial technology matter?

**Tarekegn Tariku Ebissa**[1☯*], **Deresse Mersha Lakew**[2☯]

1 Department of Accounting and Finance, Wallaga University, Nekemte, Ethiopia, 2 Department of Public Financial Management and Accounting, Ethiopian Civil Service University, Addis Ababa, Ethiopia

☯ These authors contributed equally to this work.
* tarekegnt.04@gmail.com

**Data Availability Statement:** All relevant data are within the paper and its Supporting information files.

## Abstract

This study aims to assess the effects of financial technology on the cost efficiency of commercial banks in Ethiopia. Secondary panel data were collected from the audited annual reports of seventeen commercial banks for twelve fiscal years between 2011 and 2022. The cost efficiency of banks was investigated via a stochastic frontier approach. The findings indicate that commercial banks operating in Ethiopia are efficient in cost management, with an 83% efficiency rate on average. In addition to financial technology, the effects of bank size, interest spread rate, management quality, foreign exchange rate, capitalization and bank ownership are significant for cost efficiency. Notably, financial technology plays a remarkable role in the cost efficiency of commercial banks in Ethiopia. The study results show the presence of a positive association between financial innovation platforms and the cost efficiency of commercial banks. Banking services delivered via card banking, mobile banking, and internet banking improve the cost efficiency of commercial banks in Ethiopia. As a strategic resource, financial innovation in banking operations enhances the cost efficiency of banks by reducing noneconomic costs and the time needed to deliver banking services. To improve their cost efficiency, commercial banks in Ethiopia are encouraged to use financial innovation platforms to deliver financial services and update the current cost management strategy intended to reduce operating costs incurred to generate and collect loans and advances and interest expenses paid to maintain deposits and other interest-bearing liabilities.

## Introduction

Commercial banks in Ethiopia are elderly financial institutions and highly dominate the financial system of the country compared with other forms of financial institutions. Banks take a larger share of total capital and are more accessible to customers than microfinance institutions and insurance companies in the country are. The total capital of commercial banks is ETB 199 billion, which is more than the reported capital for microfinance institutions (ETB 58.9 billion) and insurance companies (ETB 13.4 billion) and more accessible for serving 11,758.30 people per branch at the end of 2021–2022 w43].

Despite their notable dominance, the efficiency of commercial banks in Ethiopia is not impressive. The overall return on assets and efficiency rate of commercial banks show a

**Funding:** The author(s) received no specific funding for this work.

**Competing interests:** The authors have declared that no competing interests exist.

declining trend over ten years of operation. The average return on assets of banks declined from 2.83% to 2.54% from 2013/14 to 2021/22, whereas the efficiency rate declined rapidly, by 59% on average, from 113% in 2013/14 to 54% in 2021/22 [1]. Furthermore, in comparison with peer countries' financial institutions, commercial banks in Ethiopia are realizing high cost–income ratios and slow real growth rates. The cost-to-income ratio has increased due to branch expansion, large investments in financial technology and high compensation expenses [2]. On the other hand, the deposit-to-GDP ratio in Ethiopia is 33%, which is below average for African banks (37%), Kenya (37%) and South Africa (72%) [3, 4].

Commercial banks in Ethiopia are undergoing different economic reforms, financial policies and proclamation earmarks to increase their efficiency and productivity. In particular, proclamation number 1159/2019 was issued to amend the former banking business proclamation and realize digital financial services and agency banking in the financial system of the country [5]. Subsequently, card banking (automated teller machine (ATM), point-of-sale (POS) and debit card), mobile banking, internet banking and agency (tele) banking provide financial services to many commercial banks in the country. The report shows the presence of 42,416 ATMs; 84,344 POS terminals; 136.7 million debit card users; 76.3 million mobile banking subscribers; and approximately 15.7 million internet banking subscribers in the banking sector of the country at the end of 2022 [6]. However, no empirical study has investigated the effects of these digital banking platforms on the cost efficiency of commercial banks operating in Ethiopia.

The financial system in Ethiopia, predominantly the banking industry, is very vibrant. In particular, the paradigm shift in financial technology (hereafter fintech) pushes commercial banks to change the provision of financial services from traditional systems to digital banking systems. Digital banking service provisions have now become a formal tool of competition for rivals. Consequently, the use of fintech in banks and by customers shows rapid growth in the type and amount of transactions. In general, more than 2.7 billion ETB were mobilized via card banking, mobile banking and internet banking from 2010–2022 [6].

Despite the rapid shift in financial innovation in the banking industry of Ethiopia, little is known about the banking sector's cost efficiency. Banking efficiency-related empirical studies are unable to enrich the existing banking literature because of theoretical, empirical and research approach gaps. For instance, [7] assessed determinants of commercial bank efficiency in Ethiopia via data envelopment analysis (DEA). The study has employed a nonparametric approach and fails to analyze the study's findings from efficiency theory points of view.

Furthermore, [8] has examined the cost and technical efficiency of Islamic window banking in operation under the control of conventional banks for the period 2016–2020 via the stochastic frontier approach (SFA). Although parametric data analysis was applied, the study could not address the cost efficiency of other conventional banks, and the findings lack generalizability to those banks not delivering Islamic banking services.

To the best of our knowledge, adequate empirical studies are not available to assess the cost efficiency of commercial banks in Ethiopia, particularly from the fintech perspective. Furthermore, empirical studies conducted in the economic zones of Sub-Saharan Africa and East Africa are limited to considering commercial banks from Ethiopia. For example, commercial banks from Ethiopia were not considered in examining the cost and profit efficiency of commercial banks operating in Sub-Saharan Africa [9, 10] or East Africa [11]. This may be due to a lack of access to sufficient financial data regarding Ethiopian commercial banks. As a result, the literature gaps remain acute from the perspective of Ethiopian commercial banks.

Thus, this study has many empirical contributions and policy implications for the Ethiopian banking industry. First, the status and determinants of the cost efficiency of banks were assessed. Second, the effects of financial technology (fintech) on commercial banks' cost

efficiency were addressed. Third, unlike former empirical studies, a parametric approach, i.e., the stochastic frontier approach, will be fully applied. Finally, policy suggestions are presented regarding the efficient usage of deployed inputs (resources), financial technology platforms and managerial efforts in providing financial services.

## Theoretical and empirical literature review

### Adoption of financial technology by commercial banks

Commercial banks operating in developing and developed countries are adopting financial technology to facilitate and advance their banking operations. Fintech represents a way of providing automated financial services to end users via innovative technology and online electronic platforms by replacing traditional ways of transacting and marketing financial services in financial institutions. Commercial banks in Ethiopia are also adopting different fintech platforms that that allow customers to access financial services without distance and time barriers [12]. Fintech development in Ethiopia is improving customers' financial service satisfaction [13], though, its acceptance is subject to customers' awareness, ease of use, safety, attitudes and perceived usefulness [12, 14, 15].

Despite the challenges encountering financial services digitalization in Ethiopia, a rapid growth is recognized in fintech adoption by the financial sector of the country [16]. Similarly, [17] suggested that ATM deployment by Ethiopian commercial banks is rising significantly following the growth in bank size, profitability and deposit level; however, cost efficiency and bank branch expansion do not matter for the ATM deployment. Moreover, a study conducted on the impact of digital transformation on financial stability has suggested the presence of negative association between fintech adoption and financial stability of banking system in Ethiopia [18].

Empirical literature from other emerging economy is confirmed that fintech development adversely affects financial stability of banks [19]. In contrast, some scholars argue that fintech adoption boots financial stability [20] and economic growth [21] by enhancing financial inclusion [22] and financial performance of banks [23].

On the other hand, the findings of some studies argue the absence of strong empirical evidence regarding the significance of ATM transactions [24, 25] and information communication technology (ICT) expenditures [26] in the efficiency of commercial banks. Thus, controversial empirical analyses that not assessed from Ethiopian commercial banks' perspectives are reflected in the literature.

Furthermore, notable empirical findings that addressed the effects of financial technology (Fintech) on the efficiency and productivity of firms are presented in the literature. Fintech forces banks to change their business ideology in adopting new technology [27] and focusing on efficiency improvement in response to changes in the financial system [28]. In recent empirical studies, the shift in financial service provision because of fintech has fundamentally improved financial stability [29] and operational efficiency [28] and reduced the risk-taking of commercial banks [30].

Related empirical studies have also suggested significant improvements in the cost efficiency of commercial banks for those adopting electronic payments [24], IT-based transactions [31] and fintech-innovated platforms [32]. Innovation in fintech encourages the efficiency of commercial banks by reducing operating costs in borrower screening and monitoring, enhancing market service innovations [32], and strengthening quality service delivery [31]. However, some scholars have criticized the effects of fintech development in the financial industry from the viewpoints of its disruptive capability [33], demanding prudential

regulation, barrier to new entry, vulnerability to cyber-attack and high operational risk [34, 35], and it has no real cost reduction advantage in bank operations [36].

In the banking literature, cost efficiency is described as the ability of a bank to use fewer inputs to produce the same level of outputs relative to other best performing banks [37]. In our case, a bank is cost efficient when it is able to present the intended banking services at the minimum possible cost relative to the cost incurred by the best-practiced bank to provide a similar level of banking services in the Ethiopian banking industry.

Different theories intended to describe the associations between a firm's efficiency and causal factors are presented in the efficiency literature. The theory of the resource-based view (RBV) explains the effects of strategic resources on firms' financial performance and competitive advantage. According to this theory, a firm can maintain its competitive advantage and sustainable financial performance by creating unique resources and organizational capabilities that are uncommon, inimitable, and difficult to substitute in the environment in which the firm operates [38, 39].

The findings of many empirical studies are consistent with RBV theory. Empirical work to examine the effect of information technology (IT) on firm performance has concluded that technology resources enhance the financial performance of a firm through improving its efficiency [39]. A related study that applied a resource-based perspective suggested the existence of a positive association between high IT capability and the profitability of firms [40]. In the banking industry, fintech is set as a strategic resource intended to contribute to the efficiency of banks by reducing the cost and time of financial service delivery.

The bad luck hypothesis has also emerged in the banking literature with a new view that better describes the resource deployment and cost efficiency of firms. The bad luck hypothesis holds that increasing NPLs weakens the cost efficiency of commercial banks for requesting additional resource and management effort deployment to deal with default borrowers, collateral revaluation and disposal, protecting the quality of existing loans and maintaining the safety of bank records as per stakeholders' interests. This additional resource deployment increases the operating costs of a bank and subsequently deteriorates the cost efficiency of the bank [41–43]. In their empirical study, [44] suggested that an increase in nonperforming loans deteriorates the cost efficiency of banks by increasing the additional operating costs incurred for the desire to monitor delinquent and existing loans. Similarly, fintech is a new financial innovation system that requests additional resource deployment information about capital intended to support the efficiency of financial institutions.

In addition to the effects of fintech on the cost efficiency of commercial banks in Ethiopia, the effects of country-specific and bank-specific factors are addressed in this study by introducing them as a controlling variable. The inflation rate and foreign currency exchange rate are addressed as country-specific variables, and bank size, interest spread rate, capitalization, number of boards of directors, management quality and bank ownership are considered as a bank-specific controlling variables.

Existing empirical studies present notable findings regarding the associations of the aforementioned controlling variables with the cost efficiency of banks. According to [45, 46], a high inflation rate may reduce the efficiency of banks by increasing the operating costs incurred to manage bad debts, which leads to lower returns if a bank is unable to adjust its interest rate in proportion to the inflationary environment. The effect of the inflation rate on the cost efficiency of banks is determined on the basis of an action taken by the banks in response to the change. In contrast, [47] noted that the inflation rate in New Zealand enhances the cost efficiency of commercial banks operating in the country.

With respect to the foreign currency exchange rate, [48] revealed weak evidence for a positive association between the exchange rate and the cost efficiency of banks, and [47] suggested

that the impact of the foreign exchange rate was not steady across the efficiency of commercial banks in New Zealand because of the existence of distorted fluctuations in foreign exchange transactions.

Although [49, 50] remark that bank size does not determine the cost efficiency of commercial banks, [10] concludes that medium and larger commercial banks are more cost efficient than small banks in SSA. Studies conducted in SSA [51] and Ghana [52] suggest that highly capitalized banks are more cost efficient than less capitalized banks are. Likewise, an empirical study from Asian countries revealed a positive association between capitalization and the cost efficiency of commercial banks [53–55]. However, the empirical findings of [9, 47] suggest the presence of a negative association between capitalization and the cost efficiency of commercial banks.

According to [50], closely supervised interest rates improve the efficiency of commercial banks in the USA and Canada. In addition, [56] suggested that greater financial liberalization in setting interest rates results in better cost efficiency for commercial banks in East Asian countries. With respect to the impact of the number of boards of directors, empirical work from the European banking system [57] confirms that both board size and board composition significantly deteriorate the cost efficiency of banks; however, [58] reported that the cost efficiency of commercial banks in the UK increases significantly with board size and a larger non-executive board composition.

Furthermore, a significant variation in efficiency level is also observed between commercial banks owned by the government and those owned by private operators. Banks operating under the control of the government are more subject to agency costs than those operating under the control of private operators [9]. Agency costs exacerbate the inefficiency of banks.

An empirical analysis of the efficiency of commercial banks in SSA concluded that managerial activities erode the cost efficiency of banks [10]. In contrast, [55, 59] confirm that management quality enhances the cost efficiency of commercial banks operating in India and Southeast Asia.

## Materials and methods

### Sampling and data sources

This study aims to investigate the impact of financial technology on the cost efficiency of commercial banks in Ethiopia. According to the [6] report, 30 commercial banks had received banking business licenses to operate in the Ethiopian banking industry by the end of 2022. Seventeen (17) commercial banks were purposively sampled to address the impacts of financial technology on the cost efficiency of commercial banks in Ethiopia. Sampled commercial banks were Awash bank, Commercial bank of Ethiopia, Dashen bank, Wegagen bank, Hibret bank, Nib international bank, Cooperative bank of Oromia, Lion international bank, Zemen bank, Oromia bank, Buna international bank, Abay bank, Addis bank, Debub Global bank, Enat bank, Berhan international bank, Bank of Abyssinia. A commercial bank issuing audited financial statements for five consecutive fiscal years between 2011 and 2022 was considered in this study. However, the National Bank of Ethiopia (NBE), Development Bank of Ethiopia (DBE) and full-fledged interest-free banks were not included in the sample.

Secondary panel data were collected from the audited annual reports of sampled commercial banks for twelve fiscal years between 2011 and 2022. Data obtained from the audited financial statements of the sampled commercial banks were used to represent the bank-specific variables. Country-specific data were retrieved from the Ethiopian Central Statistics Agency (CSA) and the World Bank group databases. The data designate selected macroeconomic variables at the country level. Thus, the penal dataset was obtained from four data provider

entities: the National Bank of Ethiopia, the Central Statistics Agency, the World Bank group and the sampled commercial banks.

## Variable definition and measurement

The variables used in this study were grouped as dependent variables (total cost), input prices, output values, financial technology, and firm-specific and country-specific factors. The detailed definitions and measurements for each variable are presented in Table 1, with essential remarks.

## Model specification

In this study, the stochastic frontier approach (SFA) proposed by [63] is adopted to formulate a cost efficiency function for commercial banks in Ethiopia. The SFA is a parametric approach that has many advantages over nonparametric techniques, such as data envelopment analysis (DEA) and full disposable hull analysis (FDHA) in efficiency score estimation [60]. First, SFA enables the separation of inefficiency from actual operation and inefficiency due to other stochastic shocks, such as measurement error, bad luck and/or stochastic shocks [64].

Second, SFA allows the inclusion of control variables in the efficiency estimation model and provides a good comparison basis for efficiency between countries or firms [60]. Third, according to [64], the parametric technique yields a higher mean efficiency estimate, which implies that it accepts less dispersion in efficiency result estimation than the nonparametric technique does. This finding has been empirically tested and confirmed in most recent studies

**Table 1. Definition and measurement of variables.**

| Description | Variable | Notation | Definition |
|---|---|---|---|
| Dependent variables | Total cost | TC | Interest expense + personnel expenses + other administration expenses + other operating expenses [47, 60]. |
| Output quantity | Loans | $O_1$ | Total loans and advances [47, 60, 61]. |
| | Other earning assets | $O_2$ | Investment asset + securities + other earning assets [60–62]. |
| | Noninterest income | $O_3$ | Commission, exchange, brokerage, etc. [61] |
| Input prices | Price of labor | $P_1$ | Total personnel expense/total number of employees [47, 62]. |
| | Price of funds | $P_2$ | Total Interest expenses/Total deposits and other interest bearing liabilities [47, 61, 62]. |
| | Price of physical capital | $P_3$ | Operating cost–personnel cost/total fixed assets [60–62]. |
| Fixed net-put | Equity | E | Equity capital [62]. |
| Control variable | Time trend | T | Dummy as 1 for 2011 . . .. 12 for 2022. |
| Bank specific variable | Bank size | bsize | Annual total assets of a bank |
| | Interest rate spread | inter | Lending interest rate–deposit interest rate |
| | Capitalization | capit | Equity capital to total asset |
| | Board size | bosize | Number of board of director member |
| | Board composition | bcomp | Percentage of women in the board |
| | Bank ownership | bowner | 1 "government", 0 "otherwise" |
| | Management quality | mgmtq | Total operating expenses to total assets |
| Country specific variables | Inflation rate | infl | $(CPI_t − CPI_{t−1})/CPI_{t−1}$ |
| | Foreign exchange rate | fxrate | Average value of real exchange rate of ETB against US dollar for each year. |
| Financial technology | Card banking | card | Annual transaction amount via ATM and POS terminals |
| | Mobile banking | mobile | Annual transaction amount via mobile |
| | Internet banking | internet | Annual transaction amount via internet |

[25, 65]. However, the SFA is criticized for the functional form specification and normal distribution assumptions imposed on efficient frontier models. According to [38], functional form misspecifications are subject to inaccurate efficiency score estimation.

As presented in this section, the translog frontier function for cost efficiency estimation is derived from the total cost of the sampled commercial banks. The total cost (TC) efficiency measures the minimum possible input cost incurred by a firm to achieve the targeted maximum output relative to its best performer [66]. The TC function associates the price incurred to use inputs–such as labor, funds, and physical capital to produce outputs—such as loans, other earning assets (OEA) and noninterest income.

$$TC_{it} = f(P_{it}) + \varepsilon_{it} \tag{1}$$

$$\varepsilon_{it} = v_{it} + u_{it} \tag{2}$$

The stochastic TC function for $bank_i$ across time period $t$ is defined in terms of the explanatory variables $P_{it}$ and the disturbance term $\varepsilon_{it}$. $\varepsilon_{it}$ is the random disturbance term, which is assumed to encompass the random error $v_{it}$ and the inefficiency term $u_{it}$. $v_{it}$ captures random disturbance terms that incorporate the effects of measurement errors, bad luck and stochastic shocks and is assumed to follow a symmetric normal distribution around the frontier. Thus, $v_{it}$ is assumed to be independently and identically distributed with $v_{it}N(0, \sigma_v^2)$. The second error term $u_{it}$ captures managerial inefficiency terms and is assumed to follow a nonnegative truncated or half-normal distribution above the cost frontier or below the profit frontier [47] with mean $\mu$ and variance $\sigma_\mu^2$ that vary [60, 63]. That is, $u_{it}N^{+(z_{it}\mu, \sigma_\mu^2)}$.

On the other hand, the maximum likelihood estimation method is applied in estimating the parameters of the cost frontier expressed in terms of $\sigma_u^2$ and $\sigma_v^2$. The interaction between these parameters is further defined as $\sigma_T^2 = \sigma_u^2 + \sigma_v^2$; $\gamma = \frac{\sigma_u^2}{\sigma_T^2}$. The value of $\gamma$ lies between zero and one, and a larger inefficiency exists as it approaches one. It is appropriate to apply ordinary least squares when $\gamma$ approaches zero (little inefficiency exists), but it requires the use of the stochastic frontier when greater inefficiency exists to estimate the frontier model [60]. Moreover, it is possible to estimate the cost efficiency (CE) score of any bank at time $t_i$ via the following equation:

$$CE_{it} = exp(-u_{it}) = 1/exp(u_{it}) \tag{3}$$

The cost efficiency value runs between zero and one. Values closer to one imply greater cost efficiency [60, 63].

In addition, the translog frontier specification is applied in this study to estimate the parameters of cost efficiency explanatory variables, and the model takes the following forms.

$$
\begin{aligned}
\ln\left(\frac{TC_{it}}{P_3 E}\right) &= \alpha_0 + \sum_{i=1}^{2} \beta_i \ln\left(\frac{P_i}{P_3}\right) + \frac{1}{2}\sum_{i=1}^{2}\sum_{j=1}^{2} \beta_{ij} \ln\left(\frac{P_i}{P_3}\right)\ln\left(\frac{P_j}{P_3}\right) + \sum_{s=1}^{3} \gamma_s \ln\left(\frac{O_s}{E}\right) + \frac{1}{2}\sum_{s=1}^{3}\sum_{t=1}^{3} \gamma_{s,t} \ln\left(\frac{O_s}{E}\right)\ln\left(\frac{O_t}{E}\right) \\
&\quad + \sum_{i=1}^{2}\sum_{s=1}^{3} \Psi_{is} \ln\left(\frac{P_i}{P_3}\right)\ln\left(\frac{O_s}{E}\right) + \tau_1 T + \tau_2 \frac{1}{2}(T)^2 + \sum_{i=1}^{2} \emptyset_i \ln\left(\frac{P_i}{P_3}\right)*T + \sum_{s=1}^{3} \rho_s \ln\left(\frac{O_s}{E}\right)*T \\
&\quad + v_{it} + u_{it}
\end{aligned}
\tag{4}
$$

where the subscripts i denote bank, t time, ln natural logarithm of the corresponding variable, TC total cost, $P_i$ input prices, $O_i$ output quantities, E fixed net-put (equity capital), and the T time trend used to capture technical changes. $\alpha, \beta, \gamma, \tau, \Psi, \emptyset, \rho$ represents the parameters to be

estimated; $v_{it}$ represents the disturbance term; and $u_{it}$ represents the inefficiency term. In addition, the following actions are taken to ensure estimation consistency.

To hold the linear homogeneity conditions, normalization is imposed on TC, the input price of labor ($P_1$) and funds ($P_2$) using the input price of physical capital ($P_3$), whereas output quantity vectors are normalized using the fixed net-put equity capital (E) before taking the natural logarithm effects. Normalization of the dependent variables and output quantities helps prevent estimation bias that may occur due to heteroskedasticity and high economic scale differences and adds meaningful economic fact interpretation to the model [61, 67, 68].

The maximum likelihood estimation technique was subsequently applied to explore the determinants of cost efficiency on the basis of proposed explanatory variables from financial technology and controlling variables (see the detailed description in Table 1). The following econometric models were developed on the basis of the work of [63] to explain the effects of the fintech variable via the STATA software package.

$$
\begin{aligned}
CE_{it} &= \alpha_0 + \alpha_1 lncard_{it} + \alpha_2 lnbsize_{it} + \alpha_3 lninter_{it} + \alpha_4 capit_{it} + \alpha_5 BoD_{it} \\
&\quad + \alpha_6 mgmtq_{it} + \alpha_7 bowner_{it} + \alpha_8 lnfxrate_{it} + \alpha_9 lninfl_{it} + \varepsilon_{it}
\end{aligned}
\tag{5a}
$$

$$
\begin{aligned}
CE_{it} &= \beta_0 + \beta_1 lnmobile_{it} + \beta_2 lnbsize_{it} + \beta_3 lninter_{it} + \beta_4 capit_{it} + \beta_5 BoD_{it} \\
&\quad + \beta_6 mgmtq_{it} + \beta_7 bowner_{it} + \beta_8 lnfxrate_{it} + \beta_9 lninfl_{it} + \varepsilon_{it}
\end{aligned}
\tag{5b}
$$

$$
\begin{aligned}
CE_{it} &= \gamma_0 + \gamma_1 lninternet_{it} + \gamma_2 lnbsize_{it} + \gamma_3 lninter_{it} + \gamma_4 capit_{it} + \gamma_5 BoD_{it} \\
&\quad + \gamma_6 mgmtq_{it} + \gamma_7 bowner_{it} + \gamma_8 lnfxrate_{it} + \gamma_9 lninfl_{it} + \varepsilon_{it}
\end{aligned}
\tag{5c}
$$

where $CE_{it}$ represents the cost efficiency score, ln is the natural logarithm function, $\alpha_{it}, \beta_{it}, \gamma_{it}$ represents unknown parameters to be estimated for corresponding explanatory variables, and $\varepsilon_{it}$ is a disturbance term that has a truncated normal distribution, that is, $\varepsilon_{it} \approx N\left(0, \sigma_\varepsilon^2\right)$.

## Results and discussion

### Attributes of the cost efficiency of commercial banks in Ethiopia

A stochastic frontier approach is employed to assess the attributes and roles of financial technology in the cost efficiency of commercial banks in Ethiopia. An unbalanced penal dataset was obtained from the audited annual reports of seventeen selected commercial banks. The dataset covers twelve-year annual reports between 2011 and 2022. The results analysis and discussion presented in this section begin by describing the attributes of the cost efficiency of commercial banks on the basis of the evidence presented in Table 2. A discussion regarding the impacts of financial technology is presented in the next section.

Commercial banks in Ethiopia are highly efficient in cost management and operate above the cost efficiency frontier line on average. The overall average cost efficiency score of the banks was 83%, which implies that commercial banks operating in Ethiopia are inefficient at only 17% in cost management and that there is a vacuum for further improvement in cost efficiency. The cost efficiency scores of the banks vary between 96.85% and 68.80%, with the maximum and minimum values on average, respectively.

In ranking commercial banks on the basis of their cost efficiency scores, three commercial banks—Enat Bank, Commercial of Ethiopia (CBE) and Zemen Bank—are ranked 1st, 2nd and 3rd respectively, from Ethiopian commercial banks. The cumulative cost efficiency of commercial banks owned by private operators is 82.40%, which is below the overall cumulative cost efficiency of all commercial banks (83%) operating in Ethiopia.

**Table 2. Summary statistics for cost efficiency scores.**

| *Panel A*: *Commercial banks in rank* | Cost efficiency scores (%) | |
|---|---|---|
|  | **Mean** | **Std.Dev.** |
| Enat Bank | 96.85 | 0.000 |
| Commercial bank of Ethiopia | 90.66 | 0.000 |
| Zemen bank | 90.19 | 0.000 |
| Overall efficiency | 83.00 | 0.065 |
| *Panel B*: *Bank ownership* | | |
| Government owned bank | 90.66 | 0.000 |
| Private owned banks: Highest | 96.85 | 0.000 |
| Private owned banks: Lowest | 68.80 | 0.000 |
| Private owned banks: Average | 82.40 | 0.064 |
| Overall efficiency | 83.00 | 0.065 |
| *Panel C*: *Time trend (2011–2022)* | | |
| 2011 | 82.62 | 0.052 |
| 2012 | 83.02 | 0.053 |
| 2013 | 82.13 | 0.062 |
| 2014–2022 | 83.00 | 0.070 |
| Overall efficiency | 83.00 | 0.065 |

Source: Author's computation

Both the highest and lowest cost efficiency score banks are found in privately owned commercial banks. The overall cost efficiency of commercial banks owned by private operators (82.40%) is far less than that of commercial banks owned by the government (90.66%). This implies that a greater number of commercial banks owned by private operators are less efficient in cost management than government-owned commercial banks (i.e., CBE), except for Enat Bank and Zemen Bank.

The cost efficiency of Ethiopian commercial banks shows stagnant changes over the study period. The cost efficiency score of the banks was 82.62% during 2011 and nearly 83% during 2022. This implies the absence of significant changes over twelve years of bank operation in Ethiopia. Specifically, commercial banks in Ethiopia have recognized stationary cost efficiency trends in the last nine years of bank operation from 2014–2022. This result has good managerial and strategic implications.

First, it may imply that commercial banks in Ethiopia are applying similar cost management policies and strategies over the study period, particularly from 2014–2022. The banks might have ignored revising their cost management policy and strategy in line with enhancing cost efficiency. Second, there might be inflationary conditions that increase the operational costs of banks and hinder their ability to improve their cost efficiency, even though policy and strategy revisions are in place. The good news in this case is that the banks could keep their cost efficiency level at a stationary trend. Third, the cost management policies and strategies in place do not significantly support the cost efficiency of commercial banks in Ethiopia. In particular, commercial banks operating under private ownership should assess the effectiveness of cost management policies and strategies in place and revise them accordingly.

## Stochastic frontier model: Cost efficiency of commercial banks

The translog stochastic frontier model, which allows flexible estimation of parameters, was applied in this study. The estimation is made by regressing the price of inputs, output

quantities, time trend and interaction among these terms on the total cost of commercial banks in Ethiopia. The outcomes of the frontier regression presented in Table 3 are significant and acceptable for analysis.

First, the chi-square test of zero coefficient variation in the model is rejected at the 1% significance level ($x^2$ = 3534.92). This implies that the explanatory variables significantly explain the existing variations in the cost efficiency model and that the coefficients of the parameters are highly different from zero. Second, the value of sigma-squared ($\sigma^2$ = 0.0197, 3.812) is significant at the 1% significance level, implying that the estimate of the parameters is highly significant. Third, the estimated value of Gamma ($\gamma$ = 0.4034, 40.34%) is also highly significant at the 1% significance level, which implies that a significant amount of variation is derived from the inefficiency of commercial banks, whereas the variance due to random error is small. In addition, the coefficient of eta (η) (0.1001) is significantly different from zero, implying the presence of a significant difference between the results of the time-invariant and time-varying decay frontier models in this study. Thus, a time-varying decay frontier model is applied to understand the trend of changes across time in the cost efficiency of banks.

This section presents a detailed analysis of the stochastic frontier outcomes of cost efficiency (see Table 3). The results presented in Table 3 reveal that the price of both inputs—labor and funds—are not a significant driver of total cost in the Ethiopian banking industry. The employment cost and cost of debt do not account for a remarkable share of the total costs incurred by banks to run operations. There is no strong evidence that indicates the presence of a linear or quadratic association between input prices and the total cost.

With respect to the effects of output quantities, two items, loans and other earning assets (OEAs), significantly contribute to the total costs of commercial banks in Ethiopia. However, the effects are in opposite directions. The stochastic frontier results show the presence of a positive and significant association between loan output and total cost at the 1% significance level. The operating costs of generating loans increase the total cost of commercial banks in Ethiopia, and the practice of generating one additional unit of loan is more costly in the Ethiopian banking industry.

In contrast, the association between other earning assets and total cost is negative and significant at the 5% level. This implies that the practice of investing in securities and other financial assets enhances the cost efficiency of commercial banks in Ethiopia. Comparatively, emphasizing obtaining other earning assets has meaningful economic advantages over generating loans in commercial banks. However, the effects of other earning assets are not consistent. Maintaining OEA tends to enhance the total cost after a certain trend of association since a quadratic relationship that has a positive effect was obtained from the stochastic frontier regression.

On the other hand, the interaction between the input price and output quantity increases the total cost of operating banks. In particular, the combination of funds and loans enhances the total cost of commercial banks. There is a significant and positive association between the combination and the total cost of commercial banks. The results have two remarkable implications for the cost efficiency of commercial banks.

First, commercial banks in Ethiopia are not efficient enough in managing interest expenses paid to maintain deposits and other interest-bearing liabilities. Second, the positive association between loans and total cost is derived from banks' inefficiency in generating and collecting loans and advances. These weaknesses decrease the cost of efficiency for commercial banks by subjecting the banks to additional interest expenses, loan mobilization costs and delinquency costs due to nonperforming loans. Commercial banks operating in Ethiopia may improve their cost efficiency by implementing a cost management strategy that reduces interest

**Table 3. Time-varying stochastic frontier model.**

| Dependent variable–Total Cost($ln(TC/P_3 E)$) | | | Total cost | | |
|---|---|---|---|---|---|
| **Panel A: Input, Outputs and Cross terms** | | | | | |
| **Notation** | **Description** | **Parameter** | **Coef.** | **t value** | |
| $ln(P_1/P_3)$ | $ln(labor/PPC)$ | $\beta_1$ | 0.585 | 1.47 | |
| $ln(P_2/P_3)$ | $ln(fund/PPC)$ | $\beta_2$ | 0.557 | 1.28 | |
| $1/2ln(P_1/P_3)^2$ | $1/2ln(labor/PPC)^2$ | $\beta_3$ | -0.038 | -0.63 | |
| $1/2ln(P_2/P_3)^2$ | $1/2ln(fund/PPC)^2$ | $\beta_4$ | 0.097 | 1.56 | |
| $ln(O_1/E)$ | $ln(loans/Equity)$ | $\gamma_1$ | 0.902 | 2.72*** | |
| $ln(O_2/E)$ | $ln(OEA/Equity)$ | $\gamma_2$ | -0.350 | -2.10** | |
| $ln(O_3/E)$ | $ln(noninterest/Equity)$ | $\gamma_3$ | 0.189 | 1.03 | |
| $1/2ln(O_1/E)^2$ | $1/2ln(loans/Equity)^2$ | $\gamma_4$ | 0.010 | 0.06 | |
| $1/2ln(O_2/E)^2$ | $1/2ln(OEA/Equity)^2$ | $\gamma_5$ | 0.150 | 2.15** | |
| $1/2ln(O_3/E)^2$ | $1/2ln(non-interest/Equity)^2$ | $\gamma_6$ | -0.039 | -1.16 | |
| $ln(P_1/PPC)ln(O_1/E)$ | $ln(labor/PPC)* ln(loans/Equity)$ | $\Psi_1$ | -0.137 | -1.05 | |
| $ln(P_1/PPC)ln(O_2/E)$ | $ln(labor/PPC)* ln(OEA/Equity)$ | $\Psi_2$ | -0.129 | -1.24 | |
| $ln(P_1/PPC)ln(O_3/E)$ | $ln(labor/PPC)* ln(noninterest/Equity)$ | $\Psi_3$ | -0.010 | -0.12 | |
| $ln(P_2/PPC)ln(O_1/E)$ | $ln(fund/PPC)* ln(loans/Equity)$ | $\Psi_4$ | 0.303 | 2.22** | |
| $ln(P_2/PPC)ln(O_2/E)$ | $ln(fund/PPC)* ln(OEA/Equity)$ | $\Psi_5$ | 0.034 | 0.32 | |
| $ln(P_2/PPC)ln(O_3/E)$ | $ln(fund/PPC)* ln(noninterest/Equity)$ | $\Psi_6$ | 0.029 | 0.31 | |
| $T$ | $Time\ period$ | $\tau_1$ | -0.124 | -2.00** | |
| $1/2(T)$ | $1/2(Time\ period)$ | $\tau_2$ | -0.009 | -1.37 | |
| $ln(P_1/PPC)(T)$ | $ln(labor/PPC)*(Time\ period)$ | $\tau_3$ | -0.022 | -0.82 | |
| $ln(P_2/PPC)(T)$ | $ln(fund/PPC)*(Time\ period)$ | $\tau_4$ | -0.001 | -0.05 | |
| $ln(O_1/E))(T)$ | $ln(loans/Equity)*(Time\ period)$ | $\tau_5$ | 0.069 | 3.13*** | |
| $ln(O_2/E))(T)$ | $ln(OEA/Equity)*(Time\ period)$ | $\tau_6$ | 0.027 | 1.70* | |
| $ln(O_3/E))(T)$ | $ln(noninterest/Equity)*(Time\ period)$ | $\tau_7$ | -0.020 | -1.72* | |
| $Constant$ | $Constant$ | $\alpha_0$ | 1.297 | 2.33** | |
| Wald Chi-square | | | 3534.916 | 0.000*** | |
| Sigma squared | | | 0.0197 | 3.812*** | |
| Gamma ($\gamma$) | | | 0.4034 | 2.602*** | |
| Eta ($\eta$) | | | 0.1001 | 3.440*** | |
| Log-likelihood function | | | 131.02 | | |
| Number of observation | | | 198 | | |
| Number of group | | | 17 | | |

*** p<1%,

** p<5%,

* p<10%

Source: Authors' stochastic frontier regression outcomes

expenses, loan deployment costs and nonperforming loans. These findings are consistent with the bad luck hypothesis.

Across the time trend, there is a tendency toward a reduction in the total cost of operating commercial banks in Ethiopia. The association between the time trend and total cost of operating banks in Ethiopia is significantly negative at the 5% significance level. This implies considerable improvement in the cost efficiency of commercial banks over time. However, an increase in operating costs to mobilize loans and other earning assets is neutralized, and the observed improvement in the cost efficiency of commercial banks decreases. The stochastic

frontier results show the presence of a significant and positive association between loans ($\tau_5$ = 0.069) and other earning assets($\tau_6$ = 0.027) and total costs across time periods.

In contrast, the contribution from noninterest income is significantly negative ($\tau_5$ = -0.020) at the 10% significance level and enhances the cost efficiency of commercial banks over time. Thus, ensuring the sustainability of noninterest income and deploying an effective cost management strategy that minimizes the operating costs of loans and other earnings assets improve the cost efficiency of commercial banks in Ethiopia.

## Does financial technology (Fintech) matter?

As stated, this study is conducted to investigate the role of financial technology in the cost efficiency of commercial banks in Ethiopia. The effect of financial technology is investigated in this study via three proxies of fintech platforms that are currently functional in the banking industry of Ethiopian commercial banks. The first proxy is that card banking involves transaction values processed via ATM and POS terminals in the sampled commercial banks. The second proxy is mobile banking, and the third proxy is internet banking. As the outcomes are presented in Table 4, the effects of these fintech platforms are assessed through three regression models.

The chi-square test and log-likelihood function confirm the good fitness of all the proposed models. The models are highly efficient at the 1% significance level for explaining the existing variations in each model. The variables that are significant in the first model also remain significant in the estimation of the other model, suggesting the robustness of the study findings.

Table 4 presents regression outcomes for both financial technology platforms and other controlling variables. Bank size (total assets), interest spread rate, inflation rate, foreign exchange rate, capitalization, number of boards of directors, management quality and bank

**Table 4. Financial technology and cost efficiency of commercial banks.**

| Dependent variable: Cost Efficiency | | Model– 1: Card banking | | Model– 2: Mobile banking | | Model– 3: Internet banking | |
|---|---|---|---|---|---|---|---|
| Independent variables | Parameter | Coef | t value | Coef | t value | Coef | t value |
| Card banking | $\tau_1$ | 0.004 | 3.13*** | | | | |
| Mobile banking | $\tau_1$ | | | 0.005 | 2.71*** | | |
| Internet banking | $\tau_1$ | | | | | 0.005 | 2.82*** |
| Bank size | $\tau_2$ | -0.019 | -2.71*** | -0.019 | -2.60*** | -0.020 | -2.72*** |
| Interest spread | $\tau_3$ | -0.031 | -5.16*** | -0.030 | -4.89** | -0.030 | -4.98*** |
| Inflation rate | $\tau_4$ | -0.009 | -1.10 | -0.013 | -1.74* | -0.012 | -1.55 |
| Foreign exchange rate | $\tau_5$ | 0.093 | 3.96*** | 0.079 | 2.79** | 0.079 | 2.81*** |
| Capitalization | $\tau_6$ | 0.249 | 1.91* | 0.259 | 1.96* | 0.237 | 1.79* |
| Number of BoD | $\tau_7$ | -0.002 | -1.12 | -0.002 | -1.11 | -0.002 | -1.19 |
| Management quality | $\tau_8$ | -2.441 | -8.06*** | -2.468 | -7.80*** | -2.42 | -7.80*** |
| Bank ownership | $\tau_9$ | 0.079 | 3.54*** | 0.080 | 3.52*** | 0.081 | 3.59*** |
| Constant | $\tau_0$ | 0.871 | 13.74*** | 0.922 | 11.52** | 0.93 | 11.61** |
| Number of observations | 198 | 198 | 198 | | | | 11.61 |
| Chi-square | | 193.006*** | | 181.10*** | | 187.35*** | |
| Log likelihood function | | 320.612 | | 319.874 | | 319.976 | |

*** p<1%,

** p<5%,

* p<10%

Source: Authors' regression outcomes

ownership are introduced as controlling variables into each model, and their effects remain consistent across the models.

The regression outputs reveal the presence of a significant and negative association between three controlling variables–bank size, interest spread rate and management quality–and the cost efficiency of commercial banks in Ethiopia. The effects of these variables are highly significant at the 1% significance level and decrease the cost efficiency of commercial banks. These findings are consistent with the results of previous studies [10, 50].

However, the associations between other controlling variables, such as the foreign exchange rate, capitalization rate and bank ownership and the cost efficiency of commercial banks are positive and significant at the 1%, 10% and 1% levels, respectively. These controlling variables increase the cost efficiency of commercial banks operating in Ethiopia. Similar findings are also available in previous studies [48, 53, 54].

The main objective of our study is to investigate the materiality of financial technology in the cost efficiency of commercial banks. In line with this objective, this section presents an analysis supported by previous theories and empirical studies. Specifically, resource-based view (RBV) theory and several recent empirical studies have been considered. As predicted, financial technology plays a remarkable role in the cost efficiency of commercial banks in Ethiopia. The study results reveal a positive association between the aforementioned financial innovation platforms and the cost efficiency of commercial banks. The detailed analysis is presented below.

Card banking, which includes transaction values processed via ATM and POS terminals, is one of the financial technology platforms considered in this study. As hypothesized, the effect of card banking is positive and highly significant ($\tau_1 = 0.004$, t = 3.13) at the 1% significance level in relation to the cost efficiency of commercial banks in Ethiopia. In other words, a 1% increase in card banking transactions results in a 0.04% increase in the cost efficiency of commercial banks.

Similarly, the association between mobile banking and the cost efficiency of commercial banks in Ethiopia is positive and significant ($\tau_1 = 0.005$, t = 2.71) at the 1% significance level. This implies that transactions processed via mobile banking reduce operational costs for commercial banks. A 1% increase in mobile banking transactions saves at least 0.05% of the operational costs that would be incurred to run business transactions in commercial banks. In other words, mobile banking significantly enhances the cost efficiency of commercial banks operating in Ethiopia.

In addition to assessing the roles of card banking and mobile banking in the cost efficiency of commercial banks, the effect of internet banking is also addressed in this study. This approach has the advantage of verifying the robustness of the study findings. As expected, the study results reveal the presence of a significant and positive ($\tau_1 = 0.005$, t = 2.82) association between internet banking transactions and the cost efficiency of commercial banks at the 1% significance level. Accordingly, a 1% increase in internet banking transactions results in a 0.05% increase in the cost efficiency of commercial banks. Banking transactions processed via internet banking promotes the cost efficiency of commercial banks operating in Ethiopia.

These findings are consistent with the findings of previous studies both empirically and theoretically. Empirically, previous studies suggest that financial innovation enhances the cost efficiency of commercial banks by reducing operating and monitoring costs, enhancing market service innovation [32] and strengthening quality service delivery [31]. However, the findings contradict the findings of [25, 36] who outlined the absence of actual cost reduction due to financial innovation in operating banks.

Theoretically, the findings of our study are consistent with those of resource-based view (RBV) theory and the bad luck hypothesis. RBV theory argues that a firm can maintain its

competitive advantage and sustainable financial performance by creating unique resources that are uncommon, inimitable, and difficult to substitute in the environment in which the firm operates [39, 40, 69].

Moreover, the bad luck hypothesis suggests the presence of a positive association between the cost efficiency of banks and resource deployment [41–44]. Financial technology is a strategic resource deployed in the banking industry to reduce the operating, monitoring and evaluation costs of banking service-related performance. Fintech in banking operations enhances the cost efficiency of banks by reducing noneconomic costs and time in processing banking services.

## Conclusions and managerial implications

This study mainly assesses the effects of financial technology on the cost efficiency of commercial banks in Ethiopia. The audited annual reports of seventeen commercial banks operating in Ethiopia for the fiscal year between 2011 and 2022 were used as a source of data, and a stochastic frontier approach was followed to analyze the data. Commercial banks operating in Ethiopia are efficient at managing operating costs, although there is a possible opportunity to realize further efficiency of at least 17%. In terms of cumulative value, commercial banks owned by private operators are less efficient in cost management than banks operating under government ownership. Moreover, the cost efficiency of commercial banks in Ethiopia shows stationary trends over twelve years of banking operations. This may occur when similar cost management policies and strategies are applied for a longer time without revising, monitoring and evaluating their contributions to the cost efficiency of a bank.

The translog stochastic frontier results reveal that the outcome items derive more operating costs than the input items do. Specifically, banking activities to generate loans and invest in other earning assets are major drivers of the total operating costs of commercial banks in Ethiopia. Commercial banks' main weaknesses in managing interest expenses paid for deposits and generating and collecting loans and advances are deteriorating the cost efficiency of banks in Ethiopia. In addition, operating costs raised to mobilize loans and other earning assets have neutralized and decreased the observed improvement in the cost efficiency of commercial banks over time. However, commercial banks in Ethiopia have the opportunity to improve their current cost efficiency level by realizing sustainable earnings from noninterest activities and deploying effective cost management policies and strategies.

Likewise, bank-specific and country-specific factors play significant roles in the cost efficiency of commercial banks. In particular, when bank size, interest spread rates and management quality deteriorate the cost efficiency of commercial banks, foreign exchange rates, capitalization rates and bank ownership enhance efficiency at different significance levels.

As predicted, financial technology also plays a remarkable role in the cost efficiency of commercial banks in Ethiopia. The study results show the presence of a positive association between financial innovation platforms and the cost efficiency of commercial banks. In general, banking services delivered via card banking, mobile banking, and internet banking improve the cost efficiency of commercial banks in Ethiopia. As a strategic resource, financial innovation in banking operations enhances the cost efficiency of banks by reducing noneconomic costs and time in delivering banking services. These findings are robust both empirically and theoretically [31, 32, 39, 69].

With respect to managerial and policy implications, commercial banks currently operating in Ethiopia are expected to emphasize the following key points to further improve the current cost efficiency level. First, work more on maintaining capacity for effective management of interest expenses paid to maintain deposits and other interest-bearing liabilities. Second, the

current cost management policy and strategies require revision, monitoring and evaluation with the aim of reducing operating costs, which are specifically incurred to generate and collect loans and advances. Third, they focus on boosting earnings gained from noninterest income and other earning assets, such as investment in securities and commercial assets. Fourth, financial technology should be set as one part of strategic resources intended to enhance the cost efficiency of commercial banks by reducing noneconomic costs and bank service delivery time.

## Supporting information

**S1 Data.**
(XLSX)

## Author Contributions

**Data curation:** Tarekegn Tariku Ebissa.

**Formal analysis:** Tarekegn Tariku Ebissa.

**Methodology:** Tarekegn Tariku Ebissa.

**Project administration:** Tarekegn Tariku Ebissa.

**Resources:** Tarekegn Tariku Ebissa.

**Software:** Tarekegn Tariku Ebissa.

**Supervision:** Deresse Mersha Lakew.

**Validation:** Tarekegn Tariku Ebissa, Deresse Mersha Lakew.

**Writing – original draft:** Tarekegn Tariku Ebissa.

**Writing – review & editing:** Deresse Mersha Lakew.

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
