## [Decision Letter · Decision Letter 0]

25 Nov 2024

PONE-D-24-35072Cost Efficiency of Commercial Banks in Ethiopia: Does Financial Technology Matter?PLOS ONE

Dear Dr. Ebissa,

Thank you for submitting your manuscript to PLOS ONE. After careful consideration, we feel that it has merit but does not fully meet PLOS ONE’s publication criteria as it currently stands. Therefore, we invite you to submit a revised version of the manuscript that addresses the points raised during the review process.

This is an interesting academic piece, which can offer detailed insight into the Ethiopian banking sector, but as Rev 1 points out - and I fully agree with that opinion -, a more thorough and "solid" literature review is requested on the adoption of financial technology by commercial banks, with a focus on impact, profitability, market capitalisation, etc. In addition, I also underscore that a more detailed and convincing methodological section is of utmost importance in order to secure the output of the research. Without these I do not recommend this piece for publication, but if the author/s can improve the manuscript in this way, I suggest that it gets published. ==============================

We look forward to receiving your revised manuscript.

Kind regards,

István Tarrósy, PhD

Academic Editor

PLOS ONE

Reviewers' comments:

Reviewer's Responses to Questions

**Comments to the Author**

1. Is the manuscript technically sound, and do the data support the conclusions?

Reviewer #1: Yes

Reviewer #2: Yes

2. Has the statistical analysis been performed appropriately and rigorously? 

Reviewer #1: Yes

Reviewer #2: Yes

3. Have the authors made all data underlying the findings in their manuscript fully available?

Reviewer #1: Yes

Reviewer #2: Yes

4. Is the manuscript presented in an intelligible fashion and written in standard English?

Reviewer #1: Yes

Reviewer #2: Yes

5. Review Comments to the Author

Reviewer #1: As we have little information on the Ethiopian banking sector in general, any paper with the aim of shedding light on issues of efficiency, operational costs, and profitability triggers attention in the international literature. Also, conducting research on the impact of financial technologies in emerging markets is of crutial importance. The aims, objectives, and the methodologies used in the paper are clear and easy to follow. The recommendations are indeed to be followed by the commercial banks.

The weakest points of the paper are easy to be detected. Firstly, there is no solid literature review on the adoption of financial technology by commercial banks. A number of authors measure the impact, the effects on costs, profitability, earnings, market capitalisation, etc. It would be highly important to show the findings of these papers in order to better understand the Ethiopian room for manoeuvre in the overall transformation. It would be the best to compare the impacts in case of the advanced banking sector and in case of the emerging country banks. Secondly, the reviewer has to highlight that there is no overview of the methodologies used when measuring the impact. We have a clear view on the methods used in this paper but there is little information on what others use. With the help of a quick overview, we could get a better picture on why the authors use the method mentioned and how it differs from the methods used by others.

All in all, the reviewer has a high opinion ot scientifiv contribution of the paper but corrections are need as it has been mentioned above.

Reviewer #2: In my opinion, this is a very well researched piece of work and well written article. It is based on a solid amount of data and deals with interesting topic which is not suffuciently covered by existing literature. It also gives recommendations to how commercial banks in Ethiopia should work. Overall, I do not have any reservations and fully recommend this article for publication in PLOS One.

6. PLOS authors have the option to publish the peer review history of their article (what does this mean?). If published, this will include your full peer review and any attached files.

Reviewer #1: No

Reviewer #2: No

---

## [Author Response · Author response to Decision Letter 0]

5 Dec 2024

Date: November 30, 2024

To: PlosOne Academic Editor and Reviewers 

Subject: Authors’ Response to reviewers and editor 

Dear all, I thank you very much for your constructive comments and suggestions. I appreciate the contributions of your comments, which are remarkable in our manuscript and academic quality. This is to present responses to reviewers’ and academic editor’s comments regarding our manuscript under review with a research topic of “Cost efficiency of commercial banks in Ethiopia: Does financial technology matter?” and manuscript number PONE-D-24-35072. Our responses focus on two comments forwarded during the manuscript review.

Comment #1: “There is no solid literature review on the adoption of financial technology by commercial banks. A number of authors measure the impact, the effects on costs, profitability, earnings, market capitalization, etc.”

Author's Response #1: On the basis of the comment provided above, we made a detailed revision to the theoretical and empirical sections of the manuscript by adding a new subtopic—adoption of fintech by commercial banks—that focuses on fintech development, adoption, and its effect on economic growth, profitability (financial performance), and financial stability of banks and the challenges, particularly in emerging economies. A passage presented in this section shows a revision made to the revised manuscript section. In addition, you may refer page 4-5 of the revised manuscript that submitted separately. 

Adoption of financial technology by commercial banks 

Commercial banks operating in developing and developed countries are adopting financial technology to facilitate and advance their banking operations. Fintech represents a way of providing automated financial services to end users via innovative technology and online electronic platforms by replacing traditional ways of transacting and marketing financial services in financial institutions. Commercial banks in Ethiopia are also adopting different fintech platforms that that allow customers to access financial services without distance and time barriers [63]. Fintech development in Ethiopia is improving customers’ financial service satisfaction [64], though, its acceptance is subject to customers’ awareness, ease of use, safety, attitudes and perceived usefulness [1, 26, 63]. 

Despite the challenges encountering financial services digitalization in Ethiopia, a rapid growth is recognized in fintech adoption by the financial sector of the country [5]. Similarly, [40] suggested that ATM deployment by Ethiopian commercial banks is rising significantly following the growth in bank size, profitability and deposit level; however, cost efficiency and bank branch expansion do not matter for the ATM deployment. Moreover, a study conducted on the impact of digital transformation on financial stability has suggested the presence of negative association between fintech adoption and financial stability of banking system in Ethiopia [22].

Empirical literature from other emerging economy is confirmed that fintech development adversely affects financial stability of banks [46]. In contrast, some scholars argue that fintech adoption boots financial stability [67] and economic growth [45] by enhancing financial inclusion [58] and financial performance of banks [25]. 

On the other hand, the findings of some studies argue the absence of strong empirical evidence regarding the significance of ATM transactions [9, 10] and information communication technology (ICT) expenditures [55] in the efficiency of commercial banks. Thus, controversial empirical analyses that not assessed from Ethiopian commercial banks’ perspectives are reflected in the literature.

Furthermore, notable empirical findings that addressed the effects of financial technology (Fintech) on the efficiency and productivity of firms are presented in the literature. Fintech forces banks to change their business ideology in adopting new technology [27] and focusing on efficiency improvement in response to changes in the financial system [36]. In recent empirical studies, the shift in financial service provision because of fintech has fundamentally improved financial stability [52] and operational efficiency [36] and reduced the risk-taking of commercial banks [21].

Related empirical studies have also suggested significant improvements in the cost efficiency of commercial banks for those adopting electronic payments [9], IT-based transactions [41] and fintech-innovated platforms [35]. Innovation in fintech encourages the efficiency of commercial banks by reducing operating costs in borrower screening and monitoring, enhancing market service innovations [35], and strengthening quality service delivery [41]. However, some scholars have criticized the effects of fintech development in the financial industry from the viewpoints of its disruptive capability [60], demanding prudential regulation, barrier to new entry, vulnerability to cyber-attack and high operational risk [16, 47], and it has no real cost reduction advantage in bank operations [34]. 

Comment #2: “the is no overview of the methodologies used when measuring the impact. We have a clear view on the methods used in this paper but there is little information on what others use.”

Author's Response #2: It is obvious that we have employed stochastic frontier approach (SFA) to assess the effects of fintech in cost efficiency of commercial banks in Ethiopia. The SFA is a parametric approach that has many advantages over nonparametric techniques, such as data envelopment analysis (DEA) and full disposable hull analysis (FDHA) in efficiency score estimation. The following paragraph presents the advantages and limitations of SFA over the nonparametric approaches (DEA, FDHA). In addition, you may refer page 8-9 of the revised manuscript that submitted separately. 

First, SFA enables the separation of inefficiency from actual operation and inefficiency due to other stochastic shocks, such as measurement error, bad luck and/or stochastic shocks [13].

Second, SFA allows the inclusion of control variables in the efficiency estimation model and provides a good comparison basis for efficiency between countries or firms [53]. Third, according to [13], the parametric technique yields a higher mean efficiency estimate, which implies that it accepts less dispersion in efficiency result estimation than the nonparametric technique does. This finding has been empirically tested and confirmed in most recent studies [10, 38]. However, the SFA is criticized for the functional form specification and normal distribution assumptions imposed on efficient fron¬tier models. According to [23], functional form misspecifications are subject to inaccurate efficiency score estimation.

Best regards 

Tarekegn Tariku

Corresponding author

---

## [Editor Report · Decision Letter 1]

26 Dec 2024

Cost Efficiency of Commercial Banks in Ethiopia: Does Financial Technology Matter?

PONE-D-24-35072R1

Dear Dr. Ebissa,

We’re pleased to inform you that your manuscript has been judged scientifically suitable for publication and will be formally accepted for publication once it meets all outstanding technical requirements.

Kind regards,

István Tarrósy, PhD

Academic Editor

PLOS ONE

Additional Editor Comments (optional): It was good to see that the author dealt with the reviewers' comments in a thorough manner and replied to all with care. The manuscript has been improved to the level the journal requires to publish a research piece of this kind. I now can support the publication of the paper.
---

## [Editor Report · Acceptance letter]

2 Jan 2025

PONE-D-24-35072R1 

PLOS ONE

Dear Dr. Ebissa, 

I'm pleased to inform you that your manuscript has been deemed suitable for publication in PLOS ONE. Congratulations! Your manuscript is now being handed over to our production team.

Kind regards, 

on behalf of

Dr. István Tarrósy 

Academic Editor

PLOS ONE